# Molecular Recognition and Self-Organization in Life Phenomena Studied by a Statistical Mechanics of Molecular Liquids, the RISM/3D-RISM Theory

**DOI:** 10.3390/molecules26020271

**Published:** 2021-01-07

**Authors:** Masatake Sugita, Itaru Onishi, Masayuki Irisa, Norio Yoshida, Fumio Hirata

**Affiliations:** 1Department of Computer Science, School of Computing, Tokyo Institute of Technology, W8-76, 2-12-1, Ookayama Meguro-ku, Tokyo 152-8550, Japan; sugita@bi.c.titech.ac.jp; 2Department of Bioscience and Bioinformatics, Kyushu Institute of Technology, Iizuka, Fukuoka 820-8502, Japan; onishi@irisa-lab.bio.kyutech.ac.jp (I.O.); irisa@bio.kyutech.ac.jp (M.I.); 3Department of Chemistry, Kyushu University, Fukuoka, Fukuoka 812-8581, Japan; noriwo@chem.kyushu-univ.jp; 4Theoretical and Computational Molecular Science, Institute for Molecular Science, Okazaki, Aichi 444-8585, Japan

**Keywords:** molecular recognition, self-organization, RISM/3D-RISM theory, water, solvation, fluctuation, selective ion binding, enzymatic reaction, drug screening, protein

## Abstract

There are two molecular processes that are essential for living bodies to maintain their life: the molecular recognition, and the self-organization or self-assembly. Binding of a substrate by an enzyme is an example of the molecular recognition, while the protein folding is a good example of the self-organization process. The two processes are further governed by the other two physicochemical processes: solvation and the structural fluctuation. In the present article, the studies concerning the two molecular processes carried out by Hirata and his coworkers, based on the statistical mechanics of molecular liquids or the RISM/3D-RISM theory, are reviewed.

## 1. Introduction

There are two molecular processes that are essential for living bodies to maintain their life. Those are the molecular recognition and self-organization processes [1,2].

The molecular recognition is the process in which a biomolecule such as protein binds a ligand, or a small molecule, at its active site with non-covalent bonds. A typical example of such processes is seen in an enzymatic reaction. An enzymatic reaction proceeds essentially in three steps: (1) binding substrate (reactant) molecules at the active site, (2) experiencing a chemical reaction involving the recombination of atoms, (3) releasing product molecules from the enzyme to be ready for the next catalytic cycle. The first and third steps above are nothing but the molecular recognition and reverse processes, respectively, while the second process is an *ordinary* chemical reaction driven by electronic structure changes in reactants. A characteristic of an enzymatic reaction, which distinguishes it from ordinary chemical reactions in solution, is the formation of an enzyme–substrate complex referred to as the Michaelis–Menten complex [3]. The formation of the Michaelis–Menten complex is essentially a thermodynamic process governed by the free energy change between the bound and unbound states of a pair of protein and ligand molecules. Inclusion of the equilibrium constant of the complex formation in the rate constant of the chemical reaction features an enzymatic reaction. Although we have selected just one process, or an enzymatic reaction, as an example of the molecular recognition processes, all other functions that are performed by protein, such as transportation of water molecules and ions through cell membranes, are related in one way or the other to the molecular recognition process [4,5,6,7,8]. As such, binding of drug molecules to a target molecule is nothing but a molecular recognition process [9,10,11,12,13,14,15,16].

The molecular recognition process in a biological system is distinguished from that in gas phase in one important point, which concerns “water” or “solvation” [2,6,7,8,9,10,11,12,13,14,15,16,17]. In the usual situation, one or a few water molecules and a few small ions are bound or *recognized* at the active site of an enzyme or protein before a substrate molecule is bound. Substrate molecules are most likely solvated by water, as well. So, in order for protein to bind substrate molecules, some of the water molecules should be removed or “desolvated” from the active site. The free energy change associated with the desolvation process, referred to as “desolvation free energy,” constitutes a very important part of the free energy change associated with the formation of a Michaelis–Menten complex.

The self-organization is another important characteristic of biomolecules. Two prototypical examples of the process are (1) the formation of the cell membrane, and (2) the protein folding [1,2,17,18,19,20]. The process prepares a construction of molecules in which biomolecules function as an enzyme, a signal transducer, an inhibitor, and so on. Cell membranes are a sort of lipid bilayer constructed from several ingredients, including phospholipid as a major component [17]. The phospholipid has an amphiphilic characteristic, in which the hydrophilic head-groups consist of phosphate groups, while the hydrophobic tail-groups are alkyl chains. Those molecules align to make a membrane in such a way as head-to-head and tail-to-tail are close together. At a glance, such a configuration among lipid molecules may not be stable thermodynamically as a membrane due to the two physical causes: the electrostatic repulsion among head groups and the reduced entropy originated from the ordering of the molecules. Then, why do the lipid molecules self-assemble themselves? The short answer to the question is “water” or “solvation.” Another example of the self-organization is the protein folding. Protein changes its conformation from native state to unfolded state when the thermodynamic condition is changed. However, it recovers its unique native conformation reversibly when the thermodynamic state is returned back into the native condition. The phenomenon called “protein folding” was first found by C. Anfinsen [18,19,20]. The process is also counterintuitive, considering that the molecule should climb up the huge entropy barrier to reach a unique native conformation. Of course, there are interactions among the atoms in protein, either attractive or repulsive, and they may contribute to overcome the entropy barrier. However, it can be readily proved by performing the molecular dynamics (MD) simulation of a protein in a vacuum that it is not the case. If one performs the MD simulation of a protein in a vacuum starting from the structure picked up from the protein data bank (PDB), it will quickly collapse into an unidentified non-native conformation within few picoseconds. But then, why does a protein fold into a native conformation when the thermodynamic condition is brought into the native state? The quick answer to the question is again “water” or “solvation” [2,21]. The solvation free energy, including the electrostatic as well as the hydrophobic interactions involving water molecules, plays crucial roles for protein to fold into its native state.

There is another physicochemical process that is concerned with both the molecular recognition and self-organization, which is the *structural fluctuation* [22,23,24,25]. Under normal condition in living cells, the structure of an enzyme is in spatial as well as temporal fluctuation around its native conformation due to thermal motion. A substrate molecule may not be bound at the active site of many of conformations in a fluctuated state, since it may not be comfortable geometrically as well as energetically in terms of the free energy including that of solvation. Two popular models to describe the molecular recognition process taking the structural fluctuation into account are “induced fitting” and “conformation selection” [26,27]. “Induced fitting” sees the binding process as a temporal process in which a host molecule opens its *gate* or *mouth* consisting of amino acid residues for accommodating a guest molecule, induced by a perturbation due to the guest molecule. The conformational selection, on the other hand, interprets the binding process in terms of an ensemble of the host molecules, including those having the conformation ready for binding the guest molecule. The binding process is a stochastic process depending on the probability of a guest molecule to find the host molecule in the conformation that is *favorite* for binding. The structural fluctuation also plays a crucial role in self-organization processes in enzymatic reactions [28]. An enzyme as a catalyst should restore its original structure to complete its reaction, and be ready for the next reaction cycle [29]. The process is nothing but a self-organization process. Without this process, the protein would stay in a structure that does not function as an enzyme, and it loses its activity as a catalyst. 

In the present paper, the theoretical as well as computational studies concerning the molecular recognition and self-organization processes in life phenomena, carried out by authors’ group, are reviewed. The theory requires a treatment of water in molecular detail as was implied in the previous paragraphs. For that purpose, we have employed the statistical mechanics of molecular liquids, or the XRISM and 3D-RISM theories, developed based on the RISM theory originated by Chandler and Andersen [2,21,30,31,32,33,34,35].

## 2. Brief Review of the 3D-RISM/RISM Theory

Let us begin the section with asking the following questions to the readers. “What is the structure of liquid?” “How can the structure of liquid be characterized?” These questions are non-trivial, because unlike molecules and crystal, the liquid state does not form a structure of definite shape. One can readily define the structure of a molecule by giving the bond lengths, bond angles, and dihedral angles even for the most complex molecule like protein. The crystalline structure of solid can be also defined unambiguously by giving the lattice constants. However, molecules in liquids are in continuous diffusive motion, and thereby the definite geometry among the molecules cannot be defined. In such a case, we can only use the statistical or probabilistic language [2].

The probabilistic language to characterize the structure of liquids is the distribution functions, which are nothing but the moments of the density field, $\nu (r)=\sum_{i}\delta (r-r_{i})$, with respect to the Boltzmann weight. If there is no field applied to the system, the first moment or the average density is just constant everywhere in the system, namely, ρ(r)≡〈ν(r)〉=ρ=N/V where *V* and *N* are the volume of the container and the number of molecules in the system, respectively, and 〈⋅⋅⋅〉 indicates the thermal average. So, the average density does not convey any information with respect to the liquid. However, if you look at the second moment, $\rho (r,r^{\prime })=\langle \nu (r)\nu (r^{\prime })\rangle$, this quantity carries the structural information of liquids. The quantity is referred to as the density pair distribution function, which has essentially the same physical meaning as the radial distribution function (RDF) obtained from X-ray diffraction measurement. The density pair distribution function $\rho (r,r^{\prime })$ is proportional to the probability density of finding two molecules at the two positions **r** and **r’** at the same time, and it becomes just a product of the average density when the distance of the two position becomes so large that there is no “correlation” between the density of the two positions as in Equation (1).
(1)$$\underset{\left|r-r^{\prime }\right|\to \infty }{\lim}\rho (r,r^{\prime })\to \rho (r)\rho (r^{\prime })\;(=\rho^{2}\;\mathrm{in}\;\mathrm{uniform}\;\mathrm{liquids})$$

The quantity $g(r,r^{\prime })=\rho (r,r^{\prime })/\rho^{2}$ represents a “correlation” of the density at the two positions **r** and **r’**. So, it is referred to as the pair correlation functions (PCFs), or the radial distribution functions when the liquid density is uniform and the translational invariance is implied. We further define a function called the “total correlation function” by $h(r,r^{\prime })=g(r,r^{\prime })-1$, which represents the correlation of the density “fluctuations” at the two positions **r** and **r’** (Equation (2)),
(2)$$h(r,r^{\prime })=\langle \delta \nu (r)\delta \nu (r^{\prime })\rangle /\rho^{2}$$
where δν(r)(=ν(r)−ρ) denotes the density fluctuation. The main task of the liquid state theory is to find an equation which governs the function $g(r,r^{\prime })$ or $h(r,r^{\prime })$ based on the statistical mechanics, and to solve the equation.

As is briefly described in the Introduction, an “exact” equation referred to as the Ornstein–Zernike equation, which relates $h(r,r^{\prime })$ with another correlation function called the direct correlation function $c(r,r^{\prime })$, can be “derived” from the grand canonical partition function by means of the functional derivatives. Our theory to describe the molecular recognition starts from the Ornstein–Zernike equation generalized to a solution of polyatomic molecules, or the molecular Ornstein–Zernike (MOZ) equation (Equation (3)) [36]:(3)h(1,2)=c(1,2)+∫c(1,3)ρh(3,2)d(3)
where *h*(1,2) and *c*(1,2) are the total and direct correlation functions, respectively, and the numbers in the parenthesis represent the coordinates of molecules in the liquid system, including both the position **R** and the orientation **Ω**. The boldface letters of the correlation functions indicate that they are matrices consisting of the elements labeled by the species in the solution. In the simple case of a binary mixture, the equation can be written down labeling the solute by “*u*” and solvent by “*v*” as in Equations (4) and (5). (It is straightforward to generalize the equations to the multi-component mixtures.)
(4)$$h_{vv}(1,2)=c_{vv}(1,2)+\int c_{vv}(1,3)\rho_{v}h_{vv}(3,2)d(3)+\int c_{vu}(1,3)\rho_{u}h_{uv}(3,2)d(3)$$
(5)$$h_{uv}(1,2)=c_{uv}(1,2)+\int c_{uv}(1,3)\rho_{v}h_{vv}(3,2)d(3)+\int c_{uu}(1,3)\rho_{u}h_{uv}(3,2)d(3)$$
(6)$$h_{uu}(1,2)=c_{uu}(1,2)+\int c_{uv}(1,3)\rho_{v}h_{vu}(3,2)d(3)+\int c_{uu}(1,3)\rho_{u}h_{uu}(3,2)d(3)$$

By taking the limit of infinite dilution ($\rho_{u}\to 0$), one gets Equations (7) and (8),
(7)$$h_{vv}(1,2)=c_{vv}(1,2)+\int c_{vv}(1,3)\rho_{v}h_{vv}(3,2)d(3)$$
(8)$$h_{uv}(1,2)=c_{uv}(1,2)+\int c_{uv}(1,3)\rho_{v}h_{vv}(3,2)d(3)$$

The equations depend essentially on six coordinates in the Cartesian space, and it includes a sixfold integral. This integral is the one that prevents the theory from applications to polyatomic molecules. It is the interaction site model and the RISM approximation proposed by Chandler and Andersen [35] that enabled one to solve the equations. The idea behind the model is to project the functions onto the one-dimensional space along the distance between the interaction sites, usually placed at the center of atoms, by taking the statistical average over the angular coordinates of molecules with fixing the separation between a pair of interaction site (Equation (9)).
(9)$$f_{\alpha \gamma }(r)=\frac{1}{\Omega^{2}}\int \delta \left(R_{1}+l_{1}^{\alpha }\right)\delta \left(R_{2}+l_{2}^{\gamma }-r\right)f\left(1,2\right)d(1)d(2)$$
where $l_{1}^{\alpha }$ is the vector displacement of site α in molecule *i* from the molecular center **R**_i_. It follows that $R_{1}+l_{1}^{\alpha }=r_{1}^{\alpha }$ denotes the position of site α in molecule *i*. The angular average represented by Equation (9) is called Chandler–Andersen transformation [2]. The angular average of the second terms in Equations (7) and (8) is formidable, but the RISM approximation (Equation (10))
(10)$$c(1,2)\approx \sum_{\alpha \gamma }c_{\alpha \gamma }\left(\left|r_{1}^{\alpha }-r_{2}^{\gamma }\right|\right)$$
allows one to perform the angular average to lead the RISM equation (Equation (11))
(11)ρhρ=ω×c×ω+ω×c×ρhρ
where the asterisk denotes the convolution integrals, that is expressed by Equation (12).
(12)$$f*g=\int f(r_{1},r_{3})g(r_{3},r_{2})dr_{3}$$

Hereafter, solvent density is denoted by ρ instead of $\rho_{v}$. The new function ω, which appeared in Equation (11) during its derivation, is called the “intramolecular” correlation function, which is defined for a pair of atoms *α* and *γ* in a molecule, expressed by Equation (13).
(13)$$\omega_{\alpha \gamma }(r)=\rho \delta_{\alpha \gamma }\delta (r)+(1-\delta_{\alpha \gamma })\delta (r-l_{\alpha \gamma })$$
in which $\delta_{\alpha \gamma }$ and δ(r) are the Kronecker and Dirac delta functions, respectively. By virtue of the Dirac delta function, the term $\delta (r-l_{\alpha \gamma })$ imposes a distance constraint $l_{\alpha \gamma }$ between the pair of atoms. So, giving the distance constraints to all pairs of atoms in a molecule defines the molecular structure or geometry in terms of trigonometry. This is the way the molecular structure is incorporated into the RISM theory. 

The 3D-RISM equation for the solute–solvent system at infinite dilution can be derived from Equation (8) by performing the Chandler–Andersen transformation just for the coordinate of “solvent,” not for that of solute [2,21,33,34]. The equation reads Equation (14) as,
(14)$$h_{\gamma }(r)=\sum_{\gamma^{\prime }}\left[\omega_{{\gamma \gamma }^{\prime }}^{vv}\left(\left|r-r^{\prime }\right|\right)+\rho h_{{\gamma \gamma }^{\prime }}^{vv}\left(\left|r-r^{\prime }\right|\right)\right]c_{\gamma^{\prime }}(r^{\prime })dr^{\prime }$$
where $h_{\gamma }(r)$ and $c_{\gamma^{\prime }}$$\left(r^{\prime }\right)$ are the total and direct correlation functions of site *γ* and *γ’*, respectively, of solvent molecules at two positions **r** and **r**’ in the Cartesian coordinate, the origin of which is placed at an arbitrary position, usually inside the protein. The functions $\omega_{\gamma^{\prime }\gamma }^{vv}$(r) and $h_{\gamma^{\prime }\gamma }^{vv}$(r) are the correlation functions for solvent molecules, which appear in Equation (11). These equations can be applied to the molecular recognition process. If one views the solute molecule as a “source of external force” exerted on solvent molecules, then ρg(r)=(ρg(r)+ρ) is identified as the density distribution of solvent molecules in the “external force.” This identification called “Percus trick” is the key concept that made the formulation of the molecular recognition process possible by means of statistical mechanics [36].

The equations described above contain two unknown functions, *h*(**r**) and *c*(**r**). Therefore, they are not closed without another equation that relates the two functions. Several approximations have been proposed for the closure relations: HNC, PY, MSA, and so on [36]. The HNC closure can be obtained from the diagrammatic expansion of the pair correlation functions with respect to the density and discarding a set of diagrams called the “bridge diagrams,” which have multifold integrals. It should be noted that the terms kept in the HNC closure relation still include those up to the infinite orders of the density. Alternatively, the relation has been derived from the linear response of a free energy functional to the density fluctuation created by a molecule fixed in the space within the Percus trick. The HNC closure relation reads Equation (15),
(15)$$h(r)=\exp\left(-u(r)/k_{B}T+h(r)-c(r)\right)-1$$
where $k_{B}$ and *T* are the Boltzmann constant and temperature, respectively, and u(r) the interaction potential between a pair of atoms in the system. Equation (15) is the relation that incorporates the physical and chemical characteristics of the system into the theory through u(r). The PY approximation can be obtained from the HNC relation just by linearizing the factor exp[h(r)−c(r)]. The HNC closure has been quite successful for describing the structure and thermodynamics of liquids and solutions, including water. However, the approximation is notorious in the low density regime. The drawback becomes fatal sometimes when one tries to apply the theory to associating liquid mixtures or solutions, especially of dilute concentration, because a solution of “dilute” concentration is equivalent to “low density” liquid for the minor component. In order to get rid of the problem, Kovalenko and Hirata proposed the following approximation, or the KH closure expressed by Equation (16) [33,34]:(16)$$g(r)=\{\begin{array}{ll} \exp(d(r)) & \mathrm{for}\;d(r)\leq 0\\ 1+d(r) & \mathrm{for}\;d(r)>0 \end{array}$$
where $d(r)=-u(r)/k_{B}T+h(r)-c(r)$. The approximation turns out to be quite successful even for the mixture of complex liquids.

The procedure of solving the equations consists of two steps. We first solve the RISM equation, Equation (11), for $h_{\gamma^{\prime }\gamma }^{vv}$(r) of a solvent or a mixture of solvents in cases of solutions. Then, we solve the 3D-RISM equation, Equation (14), for $h_{\gamma }(r)$ of a protein-solvent (solution) system, inserting $h_{\gamma^{\prime }\gamma }^{vv}$(r) for the solvent into Equation (14), which was calculated in the first step. Considering the definition *g*(**r**) = *h*(**r**) + 1, *g*(**r**) thus obtained is the three-dimensional distribution of solvent molecules around a protein in terms of the interaction site representation of a solvent or a mixture of solvents in cases of solutions. The so-called solvation free energy can be obtained from the distribution function through Equations (17) and (18) corresponding, respectively, to the two closure relations described above, Equation (15) and Equation (16) [33,37]:(17)$$\Delta \mu_{\mathrm{HNC}}=\rho^{v}k_{B}T\sum_{\gamma }\int dr\left[\frac{1}{2}h_{\gamma }^{uv}{\left(r\right)}^{2}-c_{\gamma }^{uv}(r)-\frac{1}{2}h_{\gamma }^{uv}(r)c_{\gamma }^{uv}(r)\right]$$
(18)$$\Delta \mu_{\mathrm{HNC}}=\rho^{v}k_{B}T\sum_{\gamma }\int dr\left[\frac{1}{2}h_{\gamma }^{uv}{\left(r\right)}^{2}\Theta \left(-h_{\gamma }^{uv}(r)\right)-c_{\gamma }^{uv}(r)-\frac{1}{2}h_{\gamma }^{uv}(r)c_{\gamma }^{uv}(r)\right]$$
where Θ denotes the Heaviside step function. The other thermodynamic quantities concerning solvation can be readily obtained from the standard thermodynamic derivative of the free energy except for the partial molar volume [38,39,40].

The partial molar volume, which is a very important quantity to probe the response of the free energy (or stability) of protein to pressure, including so-called “pressure denaturation,” is not a “canonical” thermodynamic quantity for the (*V*,*T*) ensemble, since the volume is an independent thermodynamic variable of the ensemble. The partial molar volume of protein at infinite dilution (Equation (19)) can be calculated from the Kirkwood–Buff equation [Kirkwood–Buff] generalized to the site–site representation of liquid and solutions [38,39,40],
(19)$$\bar{V}=k_{B}T\chi_{T}\left[1-\rho \sum_{\gamma }\int c_{\gamma }(r)dr\right]$$
where $\chi_{T}$ is the isothermal compressibility of pure solvent or solution, which is obtained from the site–site correlation functions of solutions. In the following, we show an application of the theory described above in order to demonstrate the robustness of the theory.

The example is the partial molar volume of protein, which can be calculated using Equation (19) from *h*(**r**), or equivalently from *c*(**r**) calculated by the 3D-RISM equation. The partial molar volume of several proteins in water, which are treated frequently in the literature, is depicted against the molecular weight in Figure 1 [40]. Also plotted in the same figure are the experimental data corresponding to the theoretical results. It can be readily seen that the theory reproduces the experimental results in quantitative manner. The reader may think that the results can be reproduced just by a simple consideration of the geometry of protein, or the exclusion volume of protein. However, it will not be the case. Why? It is because the partial molar volume is a “thermodynamic quantity,” not a “geometrical quantity.” The partial molar volume is a quantity that reflects all the solvent–solvent and solute–solvent interactions as well as all the configurations of water molecules in the system. On the other hand, the geometrical volume accounts just for the simplified (hardcore type) repulsive interaction between solute and solvent. All the other factors, such as the attractive interactions between solute and solvent and the solvent reorganization, are entirely missing. The volume changes due to the solvent reorganization are especially important for the partial molar volume of protein, because it is related to the “cavity” volume in protein. As has been well studied, a protein has many internal cavities in which water molecules can or cannot be bound. It can be explained with a simple “thought experiment” concerning the partial molar volume of protein.

The thought experiment is to dissolve a protein into water. If one puts a protein molecule into water, some of the cavities in the protein may accommodate water molecules, but others may not. If the cavity is not filled by water, then the cavity space will contribute to the partial molar volume of the protein. On the other hand, if the space accommodates water molecules as a result of solvent reorganization, it will cause a negative contribution to the entire volume of solution, and compensate the increase due to the cavity volume. This compensation is significant: if a cavity is filled by one water molecule, it causes the reduction of the volume by 18 cm^3^/mol. Therefore, if a theory is not able to take account the reorganization of water molecules induced by protein, it will fail to predict the partial molar volume. The quantitative agreement between the experimental and theoretical results shown in the figure demonstrate that the theory is capable of accounting for all the solute–solvent and solvent–solvent interactions as well as solvent reorganization induced by protein.

In the following sections, it is demonstrated how the RISM/3D-RISM theory is capable of describing the molecular recognition and reorganization processes.

## 3. Molecular Recognition in Life Phenomena

### 3.1. Recognition of Water Molecules by Protein

It is a well-documented fact that water is essential for living systems to maintain their life [2,41,42]. In order to clarify the role of water in living systems in molecular detail, many scientists in the field of X-ray and neutron diffraction measurement have been trying to determine the position and orientation of water molecules around and inside biomolecules, or protein and DNA [43,44,45]. Nevertheless, the task is not so easy even for the modern experimental technologies to determine the position of water molecules due essentially to the limited resolution of those experiments. It is because water molecules at the surface of protein are not always bound firmly to some specific sites of the biomolecules, but exchange the positions quite frequently. In fact, this flexibility and fluctuation of water molecules are essential for living systems to maintain their life. The diffraction measurement can only locate some water molecules that have significant residence time at some specific positions of the biomolecules. It was Imai et al. who broke through this difficult problem by means of the molecular theory of solvation, or the RISM/3D-RISM theory, which was described briefly in the preceding section [46]. 

Imai et al. carried out the RISM/3D-RISM calculation for a hen egg-white lysozyme immersed in water and obtained the 3D-distribution function of oxygen and hydrogen of water molecules around and inside the protein. The native 3D structure of the protein was taken from the Protein Data Bank (PDB). The protein was known to have a cavity composed of the residues from Y53 to I58 and from A82 to S91, in which four water molecules were determined by means of the X-ray diffraction measurement [47]. In the calculation, those water molecules are not included explicitly.

In Figure 2 depicted by green surfaces or spots are g(r) of oxygen atoms of water molecules using an isosurface representation, which is very similar to the electron density map obtained from the X-ray crystallography. They have drawn *g*(**r**), which is greater than a threshold value. The right, center, and left figures correspond, respectively, to *g*(**r**) > 2.0, *g*(**r**) > 4.0, and *g*(**r**) > 8.0. Since *g*(**r**) is unity in the bulk, the left figure indicates that the probability of finding those water molecules at the surface is more than twice as large, compared to the bulk water. As such, the water molecules depicted in the right figure have eight times higher probability to be found than those in the bulk. The water molecules are those bound firmly to some specific atom of the protein due to, say, the hydrogen bonds, and they are quite rare, as one can see from the figure. In this sense, the threshold values play the role of the “temperature” in the X-ray diffraction measurement: if you lower the temperature, you can observe more water molecules, which have weaker interaction with protein. The results suggest that the X-ray and neutron diffraction communities have acquired a powerful theoretical tool to analyze their data to locate the position and orientations of water molecules, since the theory also provides the distribution of hydrogen atoms of water molecules.

The results depicted in Figure 2 are what Imai et al. had expected before they actually carried out the calculation, although the results were entirely new by themselves in the history of statistical mechanics. Entirely unexpected was that they observed some peaks of water distribution in a cavity “inside” the protein, which are surrounded by the residues from Y53 to I58 and from A82 to S91. The results are shown in Figure 3. The left picture in Figure 3 shows the isosurfaces of *g*(**r**) > 8 for water oxygen (green) and hydrogen (pink) in the cavity. In the figure, only the surrounding residues are displayed, except for A82 and L83, which are located in the front side. There are four distinct peaks of water oxygen and seven distinct peaks of water hydrogen in the cavity. The spots colored by green and pink indicate water oxygen and hydrogen, respectively. From the isosurface plots, Imai et al. reconstructed the most probable model of the hydration structure. It is shown in the center of Figure 3, where the four water molecules are numbered in the order from the left. Water 1 is hydrogen bonding to the main-chain oxygen of Y53 and the main-chain nitrogen of L56. Water 2 forms hydrogen bonds to the main-chain nitrogen of I56 and the main-chain oxygen of L83, which is not drawn in the figure. Water 3 and 4 also form hydrogen bonds with protein sites, the former to the main-chain oxygen of S85 and the latter to the main-chain oxygens of A82 (not displayed) and of D87. There is also a hydrogen bond network among Water 2, 3, and 4. The peak of the hydrogen between Water 3 and 4 does not appear in the figure because it is slightly less than 8, which means the hydrogen bond is weaker or looser than the other hydrogen-bonding interactions. Although the hydroxyl group of S91 is located at the center of the four water molecules, it makes only weak interactions with them.

It is interesting to compare the hydration structure obtained by the RISM/3D-RISM theory with crystallographic water sites of X-ray structure [47]. The crystallographic water molecules in the cavity are depicted on the right of Figure 3 showing four water sites in the cavity, much as the RISM/3D-RISM theory has probed. Moreover, the water distributions obtained from the theory and experiment are very similar to each other. Thus, it is concluded that the RISM/3D-RISM theory can predict the water-binding sites with great success. This was the first occasion in the history of theoretical physics to probe a little molecule *recognized* by a cavity of a biomolecule.

When the results were published in the *JACS* Communications, the authors of the paper suggested that the method could be extended to the molecular recognition of other small ligands, including drug compounds, just by considering those molecules as a component of aqueous solution. The suggestion was criticized in an article in the Royal Society of Chemistry, saying “it is too much to extrapolate from the analysis of just one cavity in one protein and claim that the method is robust and widely applicable” [48]. The following few topics reviewed in rest of this section are answers to the criticism.

### 3.2. Noble Gas Recognized by Protein

Molecular recognition by protein, or ligand binding, is one of the most fundamental functions of protein in the biological process, as was emphasized in the Introduction section of this article, taking the substrate binding at active sites of an enzyme as an example. Another important example of ligand binding by protein is the binding of a drug compound to inhibit a protein activity. In either case, a ligand to be bound is a chemical compound that in general has a complicated molecular structure, consisting of many atoms with or without (partial) charges. In order for a ligand to be bound by a receptor, the geometrical shape and charge distribution of the ligand should be matched with those of the active site (or cavity) of the receptor. It is the idea behind the bioinformatic-based drug screening, or the *key-and-lock* concept. There is no question about that. However, it is not the entire story. The active sites or cavities of protein are not empty before a ligand comes into them. They are usually filled with some water molecules, an example of which was discussed in the preceding section. In order for a ligand to be accommodated in the cavity, one or a few water molecules should be disposed from the cavity to make a space. That process is a thermodynamic process requiring consumption of free energy, referred to as “desolvation free energy” [49,50,51,52,53]. 

The methodology described in the previous section can be applied to the process with a slight modification, and provides a powerful theoretical tool to realize the ligand binding by protein. The modification to be made is just to change the solvent from pure water to an aqueous solution containing ligand molecules. Presented in this section are the results for noble gases [54], which are the simplest models of non-polar ligands.

Shown in Figure 4 are the 3D distribution functions of xenon and water (oxygen and hydrogen) around lysozyme, calculated by the RISM/3D-RISM theory for lysozyme in a water–xenon mixture at the concentration of 0.001 M. The molecular surface of the protein is drawn with blue color. The regions of g(**r**) > 8 are painted with different colors for different species: yellow, xenon; red, water oxygen; white, water hydrogen. Of course, the surface painted with blue color is covered by water molecules weakly bound to the protein, which are not shown. A number of well-defined peaks, yellow and red spots, are found for xenon and water oxygen at the surface of the protein, which are separated from each other. The result demonstrates the great capability of the RISM/3D-RISM theory to predict the “preferential binding” of ligands. The distributions of ligand and water are simultaneously found in this result, which means that the peak of either the ligand or water is found at each site depending on the ratio of their affinities to the site. Actually, Figure 4 indicates that there are water- and xenon-preferred sites on the protein surface. Similar results were obtained for the other gases and the other concentrations.

It is of interest to compare the distribution of xenon obtained by the RISM/3D-RISM theory with the xenon sites in the X-ray structure [55], even though their conditions are not the same: the former is an aqueous solution under atmospheric pressure, while the latter is crystal under xenon gas pressure of 12 bar. There are two binding sites of xenon in lysozyme: one corresponds to the binding pocket of native ligands, which is referred to as the substrate binding site, and the other is located in a cavity inside the protein, which is referred to as the internal site. The theoretical results of the 3D distribution of xenon are compared with the X-ray xenon site at the substrate binding site in Figure 4a. The location of a high and sharp peak found by the theory is in perfect agreement with the X-ray xenon site. Shown in Figure 4b are the results at the internal site. The xenon peak found there is actually a minor one. Nevertheless, the location is again consistent with the X-ray site. It is of interest to note that the peaks of water are shifted off from the xenon binding site.

Shown in Figure 5 is the size dependence of the coordination number of noble gases at the two binding sites, which was calculated at the concentration of 0.001 M. At the substrate binding site, the coordination number becomes exponentially larger as the size of gas increases (Figure 5a). At the internal site, the coordination number becomes larger with increasing the gas size up to s ~3.4 Å, while it decreases in the region where σ>3.4 Å (Figure 5b). As a result, argon has the largest binding affinity to the internal site. These results demonstrate that the RISM/3D-RISM theory has the ability to describe ligand size selectivity in the binding, or molecular recognition. Although there are no corresponding experimental data, the present results serve as a representative test case.

### 3.3. Selective Ion-Binding by Protein

Selective ion binding by protein plays an essential role in a variety of physiological processes. The binding of calcium ions by some protein initiates the process to induce the muscle contraction and enzymatic reactions [56,57]. The initial process of the information transmission through an ion channel is an ion binding at the pore of the transporter molecule [58]. The ion binding plays an important role, sometimes even in the folding process of a protein by inducing the secondary structure [59]. Such processes feature the highly selective ion binding by proteins. Therefore, it is of great interest for the life sciences to elucidate the origin of the ion selectivity in molecular detail. For that purpose, a RISM/3D-RISM calculation was carried out for aqueous solutions of three different electrolytes, CaCl_2_, NaCl, and KCl, and for four different mutants of human lysozyme: wild-type, Q86D, A92D, Q86D/A92D, which have been studied experimentally by Kuroki and Yutani [60,61,62].

Shown in Figure 6 are the 3D-distributions of water molecules and the cations inside and around the cleft under concern, which consists of amino acid residues from Q86 to A92. The area where the distribution function, g(**r**), is greater than five is painted with a different color for each species: oxygen of water, red; Na^+^ ion, yellow; Ca^2+^ ion, orange; K^+^ ion, purple. For the wild type of protein in the aqueous solutions of all the electrolytes studied, CaCl_2_, NaCl and KCl, there are no significant distributions (g(**r**) > 5) observed for the ions inside the cleft, as is seen in the figure in the upper left. The Q86D mutant shows essentially the same behavior as that of the wild type, but with the water distribution being changed slightly (upper center figure). A trace of yellow spot is seen, which suggests a slight possibility of an Na^+^ ion bound in the middle of the binding site, but it will not be large enough to make any significant contribution to the distribution. In place, a rather large distribution that corresponds to water oxygen is observed, as is indicated with the red color in the figure. The distribution covers faithfully the region where the crystallographic water molecules were detected, as shown with the spheres colored gray. There is a small difference between the theory and the experiment. That is the crystallographic water bound to the backbone of Asp-91. The theory does not reproduce the water molecule, the reason for which is not known. Except for the small difference, the theoretical observation is in accord with the experimental finding, especially in the respect that the protein with the wild-type sequence does not bind either Na^+^ or Ca^2+^.

On the other hand, the A92D mutant in the NaCl solution exhibits distinct distributions of an Na^+^ ion bound in the recognition site, which is consistent with the finding by the experiment (upper right figure). The Na^+^ ion is bound primarily to the carbonyl oxygen atoms of Asp-92 and has the distribution around the atoms. There is a water distribution in the active site, but the form of the distribution is altered entirely from that in the wild type. The change in the distribution indicates that the Na^+^ ion bound in the active site is not naked, but is hydrated by water molecules. The mutant shows no indication of a K^+^ ion bound to the site. (The results are not shown.) It suggests that the A92D mutant is able to discriminate an Na^+^ ion from a K^+^ ion. The finding demonstrates the capability of the RISM/3D-RISM theory to realize the ion selectivity by protein. 

Shown in the lower panels are the 3D-distributions of Ca^2+^ ions and of water oxygen in the ion binding site of the holo-Q86D/A92D mutant. The mutant is well regarded experimentally as a calcium binding protein. The mutant, in fact, exhibits a strong calcium binding activity as can be seen in the figure. The calcium ion is bound by the carboxyl groups of the three Asp residues and is distributed around the oxygen atoms. The 3D-distribution of water at the center of the triangle made by the three carbonyl oxygen atoms is reduced dramatically, indicating that the Ca^2+^ ion is coordinated by the oxygen atoms directly, not with water molecules between them. The Ca^2+^ ion, however, is not entirely naked, because the persistent water distribution is seen at least at two positions where original water molecules were located in the wild type of the protein. 

The results obtained in the study gave a great confidence to the authors to apply the method to actual enzymatic reactions in which ions as cofactors play important roles. The following topics concern the role of ions in an enzymatic reaction.

### 3.4. Molecular Recognition in an Enzymatic Reaction: Role of Mg^2+^ Ions in DNA Hydrolysis Reaction by Restriction Enzyme EcoRV

It is known that small ions such as magnesium ions play an important role as “cofactors” in enzymatic reactions [63]. Moreover, their effect is extremely sensitive to the type of ion. For example, in the case of Escort, one of the type-II restriction enzymes that decompose DNA by hydrolysis reaction, the reaction does not proceed just by replacing Mg^2+^ ion with Ca^2+^ ion [64]. This suggests that the effect of the cofactor cannot be explained simply by the general physics called the “Coulomb interaction,” but is extremely specific to ionic species, the atomic detail of binding position (or distribution) in the active site, and its fluctuation. Moreover, the distribution of ions and their fluctuations are closely correlated (conjugated) with the structural fluctuations of proteins, and the binding position (or distribution) must naturally change as the reaction progresses. So far, research on hydrolysis reactions with restriction enzymes has been actively conducted to determine the positions of small ions at various stages of the reaction through techniques such as X-ray crystallography, but consensus has not yet been reached [65,66,67,68]. The main reason for this is that the structural fluctuations of the protein–DNA complex and the ion distribution are closely coupled to each other. In other words, the ion position changes significantly as the reaction proceeds.

In order to clarify the position of Mg^2+^ ions and their role in the hydrolysis reaction due to the restriction enzyme (*Eco*RV), Onishi et al. carried out a RISM/3D-RISM calculation and a molecular dynamics (MD) simulation for the precursor state of the reaction [69].

Figure 7 shows the distribution of ions in the *Eco*RV–DNA complex determined using the 3D-RISM/KH theory. In the figure, the distribution of ions is shown by the orange network structure. It is interesting to compare this result with the experimental result based on X-ray crystallography (Figure 7b) [68]. First, the ion positions, I*, II*, and III* determined by the experiment, correspond roughly to the peaks, I^†^, II^†^, III^†^ of the distribution estimated by the RISM/3D-RISM calculation. On the other hand, the theoretical result due to RISM/3D-RISM shows another peak (IV^†^) of the distribution that was not detected by X-ray crystallography. The height of this peak is about a half that of the other peaks, indicating that the position is not the most comfortable position in the equilibrium condition. It may be the reason why the ion position was not detected experimentally by the X-ray analysis. Nevertheless, it is quite unlikely that the ion at the position with such high probability is doing nothing in the reaction.

Therefore, in order to clarify the role of the ion at the position (IV^†^), an MD simulation was performed with the structure in which one Mg^2+^ ion was placed at this position as the initial structure. The result is shown in Figure 8. The initial structure of the MD simulation is shown in Figure 8a. (The initial structure also shows the arrangement of water molecules obtained by the RISM/3D-RISM calculation.) Figure 8b shows the structure after one nanosecond.

As a result of this MD simulation, the following changes took place in the structure of the *Eco*RV–DNA complex and in the arrangement of the solvent (water and ions).

The phosphate group (scissile phosphate group) involved in DNA cleavage was twisted, and moved to the position suitable for nucleophilic attack.The Mg^2+^ ion at the position IV^†^ moved to position “B” in the figure, and a water molecule serving as the substrate was placed at the proper position in the reaction.The position and orientation of one water molecule changed, and it moved to the position where it could act as a nucleophile.

In the study, the equilibrium structure obtained by the MD simulation was referred to as the M structure. Interestingly, the M structure is very similar to the structure of *Bam*HI-DNA obtained by X-ray crystallography in both the structure of the protein–DNA complex and the ion-binding position (Figure 9) [70]. In particular, the position of *Eco*RV in the M structure almost overlaps with the position of *Bam*HI (Figure 9b).

*Bam*HI is a protein similar to *Eco*RV, but Mg^2+^ is replaced by Ca^2+^ and has no catalytic activity for hydrolysis reactions. For this reason, *Bam*HI has been assigned to the structure immediately before the hydrolysis reaction, and the results of this study are in harmony with this experimental hypothesis [70].

### 3.5. Molecular Recognition in Drug Screening 

The in silico drug discovery is the most interesting field where the *molecular recognition* may play a crucial role.

The prediction of the ligand binding sites and affinities is the starting point for the drug discovery [71,72]. Therefore, a large number of computational as well as experimental approaches have been devoted to solve the problem [73,74,75,76]. The computational methodologies are classified into two categories or stages. One is the prediction of ligand binding sites in a target protein. The binding sites are found, in the most common cases, based on a purely geometric analysis of the protein structure, in which cavities or clefts in the protein are detected and regarded as the potential binding sites [73]. The binding sites can also be predicted by the bioinformatics-based methodologies such as the multiple alignment of the amino acid sequences for a protein family [77]. The other category is the docking of a ligand molecule at the binding sites that are already known or predicted in advance. Possible docking structures are then evaluated based on a force field or a scoring function [74,75,76].

Although such docking programs are increasingly popular in the fields of bioscience and pharmacology [78], the theoretical methodologies based on the physical chemistry are not fully developed. One of the least developed methodologies is how to account for the effect of water in the binding affinity or free energy. Water plays multiple roles in the binding affinity. For examples, bulk water exerts the reaction field, acting on a pair of ligand and receptor molecules. This effect includes the electrostatic screening and the hydrophobic interaction between protein and ligand molecules. Individual water molecules can act as integral molecular components of the complex [79]. In fact, water molecules are often found at the binding interface of protein–ligand complexes mediating with the hydrogen bonds or simply filling void spaces. Water around and inside a protein molecule regulates the structural fluctuation of the biomolecule, which of course has a significant effect on the binding affinity. In spite of such significance of water molecules, the effect of water has conventionally been treated at the level of continuum solvent models [74,75,76]. In so many words, it is obvious that such models will not account for the desolvation free energy properly. On the other hand, it is quite reasonable to expect that the RISM/3D-RISM theory may play a great role in this step of drug discovery. In fact, the theory has been applied to in silico drug discovery in a variety of ways [12,80,81,82]. As an example of those applications of the RISM/3D-RISM theory to drug screening, the study carried out by Hasegawa et al. is briefly reviewed, which takes “Pim1 kinase” as a target protein and “triazolopyridazine” as well as its derivatives as inhibitors [16].

The Pim1 kinase belongs to the Pim (Proviral Integration-site MulV) family along with Pim1, 2, and it is a Serine/Threonine Kinase. The 3D structure of a Pim1 kinase (PDBID: 3BGQ) with an inhibitor is shown in Figure 10. The enzyme is expressed widely in our body due to its phosphorylation activity of substrates concerning a variety of bio-functions such as cell cycle, apoptosis, and differentiation [12,83,84,85]. In particular, the enzyme is expressed excessively in malignant tumors such as leukemia, lymphoma, and prostatic carcinoma [84,85,86,87,88]. On the other hand, a mouse, the Pim1 kinase of which is knocked out, did not show any indication as a phenotype [89,90,91,92,93]. Therefore, it is considered that such a tumor may be treated by an inhibitor of the enzyme, with minor side effects. Although there have been some reports about the compounds that are tightly bound to the Pim1 kinase, they have not been commercialized yet as an actual cancer drug [89,90].

The experimental values of the inhibition constant ($K_{i}$), which is a measure of the binding affinity, observed by Grey et al. are listed in Table 1. [94]

There are two points that should be remembered when one tries to perform the rational drug screening based on the RISM/3D-RISM theory. Firstly, the criteria for screening drug candidates should be the binding affinity, which is defined by
(20)$$K_{A}=\exp\left[\frac{-\Delta G_{bind}}{k_{B}T}\right]$$
where $K_{A}$ is the equilibrium constant of the reaction,
(21)[recepter]+[ligand]⇄[complex]
and $\Delta G_{bind}$ is the binding free energy defined by
(22)$$\Delta G_{bind}=G_{com}-\left(G_{rec}+G_{lig}\right)$$
where $G_{rec}$, $G_{lig}$ and $G_{com}$ are the free energies of the receptor (protein), ligand (drug), and their complex, respectively. Unfortunately, the RISM/3D-RISM theory cannot be applied directly to calculate those free energies. However, one may take an alternative route to find $\Delta G_{bind}$ using the thermodynamic cycle depicted in the following schematic picture (Figure 11).

The binding free energy is calculated from the cycle as
(23)$$\Delta G_{bind}=\Delta G_{solute}+\Delta G_{desolv}$$
where $\Delta G_{solute}$ is the binding free energy of solutes in vacuum defined by
(24)$$\Delta G_{solute}\;=\;\Delta E_{solute}-T\Delta S_{solute}$$
in which $\Delta E_{solute}$ and $T\Delta S_{solute}$ are the energetic and entropic contributions. Those quantity can be calculated readily by means of the molecular mechanics. $\Delta G_{desolv}$ is the desolvation free energy defined by
(25)$$\Delta G_{desolv}\;=\Delta \mu_{com}-(\Delta \mu_{rec}+\Delta \mu_{lig})$$
where $\Delta \mu_{com}$, $\Delta \mu_{rec}$, and $\Delta \mu_{lig}$ are the solvation free energy of the complex, the receptor, and the ligand, respectively. Those can be calculated from the RISM/3D-RISM theory using Equations (17) and (18).

The other point of concern in the RISM/3D-RISM calculation is the structural fluctuation of protein [2]. As will be described in Section 4 in the present paper, the structure of protein is in spatial as well as temporal fluctuation, and many of the fluctuated states of protein may not bind the drug molecule to be examined. On the other hand, in the ordinary RISM/3D-RISM calculation, the structure of protein is assumed to be *rigid*, meaning that all the bond lengths, bond angles, and dihedral angles are fixed, that is, no structural fluctuation is allowed. Therefore, it is very likely that the structure (atomic coordinates) of protein picked up from the Protein Data Bank (PDB) may not be accurate enough for the ligand to be bound. In order to take the structural fluctuation into consideration, Hasegawa et al. carried out the MD simulation of the protein in water (details of the calculation are omitted here) [16].

The binding free energies of the 16 compounds to the target protein, calculated based on the MM/3D-RISM/KH method, are shown in Table 2, and plotted against the corresponding experimental values [94] in Figure 12. A rather high correlation (~0.69) is observed between the theoretical and experimental results. This result demonstrates that the MM/3D-RISM/KH method is applicable to the problem of compound screening and lead optimization, where relative affinity among the compounds has significance. However, the theoretical results show some systematic deviation from the experimental values toward the positive side. There are several conceivable causes for the systematic deviation: insufficient sampling of the conformational space of the protein, insufficient accuracy of the RISM/3D-RISM theory for estimating the solvation free energy, inadequate solution conditions, and so on. In contrast to the method that directly estimates the binding free energy, the method based on a thermodynamic cycle requires several theoretical methodologies for estimating each component of the binding free energy; inter-atomic interactions of solute, solvation free energy, conformational entropy, and external entropy. Since each method for estimating the component of the binding free energy has its own approximation, each method may systematically under- or overestimate the thermodynamic quantity. These systematic errors seem to give rise to the unphysical positive value of the binding free energy.

Among those components of the binding free energies, the conformational entropy deserves special attention because the scientists in the community of molecular dynamics simulation are experiencing a hard time to find converged results for the quantity, even after sampling a sub-micro-second length of trajectory. The situation reminds us of Levinthal’s estimate of the conformational degrees of freedom, which amounts to ~10^50^ for a small protein having ~100 amino acids [95]. It may be impossible to find the converged results for the conformational entropy by the standard method of the simulation. An analytical approach based on the RISM/3D-RISM method combined with the generalized Langevin theory described in the following section may have an advantage in that respect.

## 4. Structural Fluctuation and Reorganization of Protein in Water

### 4.1. The Theory of Structural Fluctuation of Protein

The structural fluctuation of protein coupled with the density fluctuation of solvent (water or ion) plays an essential role in its functional expression. By combining the 3D-RISM theory and the generalized Langevin theory [96], Kim and Hirata proposed a new theory that describes the structural fluctuations of protein coupled with the density fluctuations of solvent, with the purpose of constructing a computational scientific methodology [2,23].

This theory is described by the following Langevin-type equation.
(26)$$M_{\alpha }\frac{d^{2}\Delta R_{\alpha }(t)}{dt^{2}}=-k_{B}T\sum_{\beta }{\left(L^{-1}\right)}_{\alpha \beta }\cdot \Delta R_{\beta }(t)-\int_{0}^{t}\Gamma_{\alpha \beta }(t-s)\cdot \frac{P_{\beta }(s)}{M_{\beta }}+W_{\alpha }(t)$$

In Equation (26), $\Delta R_{\alpha }$ represents the fluctuation (displacement from the equilibrium structure) of the α−th atom in the protein and is defined by $\Delta R_{\alpha }(t)=R_{\alpha }(t)-\langle R_{\alpha }\rangle$, where $R_{\alpha }(t)$ and $\langle R_{\alpha }\rangle$ denote the coordinate of atom α and its ensemble average. In this equation, the first term on the right side is the restoring force (Hook type) proportional to the displacement, the second term is the memory term (friction term), and the third term is the random force originated from thermal motion. Included in the first term is the variance–covariance matrix, whose elements are defined by the following equation.
(27)$$L_{\alpha \beta }=\langle \Delta R_{\alpha }\Delta R_{\beta }\rangle$$

If one ignores the second and third terms on the right side of Equation (26), it has the form of a so-called harmonic oscillator. Therefore, Equation (26) expresses a physical picture in which the motion of this harmonic oscillator is driven by a random force resulting from thermal motion and attenuated by a frictional force proportional to the velocity. If the protein is placed in a vacuum, $k_{B}T{\left(L^{-1}\right)}_{\alpha \beta }$ in the first term of Equation (26) is a so-called force constant (Hessian), which is expressed by the second derivative of the potential energy with respect to the atomic coordinates, or displacement. However, since the actual protein is in aqueous solution, it is not just the *mechanical* potential energy of the interatomic interaction. It must be the second derivative of the *potential of mean force* (or *free energy*) including the *solvation free energy.* That is,
(28)$$k_{B}T{\left(L^{-1}\right)}_{\alpha \beta }=\frac{\partial^{2}F\left(\left\{R\right\}\right)}{\partial \Delta R_{\alpha }\Delta R_{\beta }}$$
(29)F({R})=U({R})+Δμ({R})

In Equations (28) and (29), {**R**} represents the structure (atomic coordinate) of the protein, and F({R}), U({R}) and Δμ({R}) are the potential of mean force (or free energy), potential energy, and solvation free energy, respectively. That is, the structural fluctuation of protein described by Equation (26) must be the fluctuation in the free energy surface including the solvent. The free energy surface has a quadratic form, as can be seen by performing the second integral on the displacement of the coordinates in Equation (28). However, the free energy surface is a quadric surface in a multidimensional space spanned by the coordinates of the number of atoms (~10^4^) in the protein.
(30)$$F\left(\left\{R\right\}\right)=\frac{1}{2}k_{B}T\sum_{\alpha ,\beta }\Delta R_{\alpha }\cdot {\left(L^{-1}\right)}_{\alpha \beta }\cdot \Delta R_{\beta }$$

This also means that the structural fluctuations in the free energy surface defined by Equation (30) have a Gaussian distribution. That is,
(31)$$w(\Delta R_{1},\Delta R_{2},\ldots ,\Delta R_{N})=\sqrt{\frac{\left\|A\right\|}{{\left(2\pi \right)}^{3N}}}\exp\left[-\frac{1}{2}\sum_{\alpha }\sum_{\beta }A_{\alpha \beta }\Delta R_{\alpha }\Delta R_{\beta }\right]$$
where $A_{\alpha \beta }$ is an element of the matrix expressed by the following equation, and ‖A‖ is its determinant.
(32)$$A_{\alpha \beta }=\frac{\partial^{2}F\left(\left\{R\right\}\right)}{\partial \Delta R_{\alpha }\partial \Delta R_{\beta }}$$

The physics implied by Equation (31) for the distribution of protein structural fluctuations is consistent with the experimental results of small-angle X-rays and neutron scattering [97]. If Equation (31) is transformed into Fourier space, the left side corresponds to the intensity (structural factor) of the scattering experiment, while the right side becomes a Gaussian function concerning the wave vector. Therefore, when the logarithm of both sides is taken and plotted against the square of wave vector, a straight line having a negative slope is obtained. Such a plot is called a Guinier plot in the field of small-angle X-ray scattering experiments, and Equation (31) perfectly matches such an experiment [2,97].

The expression of Equation (28) regarding Hessian, which defines the structural fluctuation of protein, gives a great perspective for describing the structural response to thermodynamic perturbation and its dynamics. This is because the solvation–free-energy surface contained in the expression can be obtained by the RISM/3D-RISM theory developed by Hirata and his coworkers [2]. The linear response theory that describes the structural response of proteins to thermodynamic perturbations such as “temperature,” “pressure,” “denaturing agents,” and “amino acid substitutions” is derived as follows. First, the change in free energy due to thermodynamic perturbation is represented by the following formula [23]:(33)$$F\left(\left\{R\right\}\right)=\frac{1}{2}k_{B}T\sum_{\alpha ,\beta }\Delta R_{\alpha }\cdot {\left(L^{-1}\right)}_{\alpha \beta }\cdot \Delta R_{\beta }-\sum_{\alpha }\Delta R_{\alpha }\cdot f_{\alpha }$$

The first term on the right side of Equation (33) is the free energy of the non-perturbed system expressed by Equation (30). On the other hand, the second term is the change in free energy due to perturbation, which is expressed by perturbation ($f_{\alpha }$) and structural displacement ($\Delta R_{\alpha }$) due to the perturbation. Applying the variational principle ($\partial F/\partial \Delta R_{\alpha }=0$) to this equation gives the following equation that expresses the structural response of the protein to the thermodynamic perturbation [23,24,98]:(34)$$\langle \Delta R_{\alpha }\rangle ={\left(k_{B}T_{0}\right)}^{-1}\sum_{\beta }{\langle \Delta R_{\alpha }\Delta R_{\beta }\rangle }_{0}\cdot f_{\beta }$$

According to Equation (34), the average structure (atomic coordinates) changes,$\langle \Delta R_{\alpha }\rangle$, responding to the thermodynamic perturbation, and its response function is the variance–covariance matrix ${\langle \Delta R_{\alpha }\Delta R_{\beta }\rangle }_{0}$ of the non-perturbed system. Equation (34) can be further extended to the “non-linear” region by applying the idea of “analytical continuation.” To do this, we first divide the perturbation ($f_{\beta }$) in Equation (34) into several steps so that the individual perturbation stays within the linear regime. Letting the *j*-th perturbation be $f_{\beta }^{j}$, the response to the perturbation is expressed by the following equation [2,24]:(35)$${\langle \Delta R_{\alpha }\rangle }_{j+1}={\left(k_{B}T_{j}\right)}^{-1}\sum_{\beta }{\langle \Delta R_{\alpha }\Delta R_{\beta }\rangle }_{j}\cdot f_{\beta }^{j}$$

The response function (${\langle \Delta R_{\alpha }\Delta R_{\beta }\rangle }_{j}$) contained in Equations (34) and (35) is related to the free energy of the protein by Equation (28), and the free energy is, in turn, given by the RISM/3D-RISM theory as a function of the structure (atomic coordinate) of the protein. Therefore, it is possible to calculate the structural response of the protein to thermodynamic perturbations [2,24].

### 4.2. Incoherent Elastic Neutron Scattering

The structural fluctuations of the protein are directly reflected in the mean square displacement, $M=\sum_{\alpha }\langle \Delta R_{\alpha }^{2}\rangle$, obtained from the incoherent scattering of neutrons. When this mean square displacement is plotted against temperature, it increases linearly with temperature, but its slope changes rapidly around 230 K. Various physical interpretations such as “glass transition” [99], “harmonic to unharmonic transition” [100], and “alpha to beta transition” [101] have been given to this behavior, but no conclusion has been reached yet.

Based on the theory summarized in the preceding section, Hirata gave a new physical interpretation of the mean square displacement of proteins and its temperature dependence [2,25].

The structure factor obtained from the neutron incoherent (elastic) scattering experiment is defined by the following equation.
(36)$$S^{EISF}(Q,\omega =0)\equiv \int w^{inc}\left(\left\{\Delta R\right\}\right)\exp\left(-iQ\cdot \Delta R\right)d\Delta R$$

The experiment is carried out by irradiating powdered material with thermal neutrons. Therefore, the material can be considered to satisfy the spatial isotropic condition as in the case of the solution system. In Equation (36), $w^{inc}\left(\left\{\Delta R\right\}\right)$ is essentially a probability distribution function of structural fluctuations defined in Equation (31), but only the hydrogen atom (H) has a large scattering cross section with respect to neutrons. Further, given that it is *incoherent* scattering, we obtain:(37)$$w^{inc}\left(\left\{\Delta R\right\}\right)={\left(2\pi L_{\alpha \alpha }\right)}^{-3n/2}\exp\left[-\frac{1}{2}\sum_{\alpha }^{n}\frac{\Delta R_{\alpha }^{2}}{L_{\alpha \alpha }}\right]$$

Here, α signifies the hydrogen atom, $L_{\alpha \alpha }$ means the diagonal term of the variance/covariance matrix, and is given by the following equation.
(38)$$L_{\alpha \alpha }=\langle \Delta R_{\alpha }^{2}\rangle$$

By substituting Equation (37) into Equation (36) and considering the “isotropicity” of fluctuations, the integration of Equation (36) can be performed analytically, and the following formula for the incoherent scattering factor of neutrons is obtained.
(39)$$S^{EISF}(Q,\omega =0)=\prod_{\alpha }^{n}\exp\left(-\frac{L_{\alpha \alpha }}{2}Q^{2}\right)$$
or taking the logarithm,
(40)$$\log S^{EISF}(Q,\omega =0)=-\frac{1}{2}\left(\sum_{\alpha }^{n}L_{\alpha \alpha }\right)Q^{2}=-\frac{1}{2}MQ^{2}$$

Equation (40) means that the incoherent neutron scattering factor decreases linearly with the square of the wave vector ($Q^{2}$), and predicts the experimental results. Furthermore, since it is intuitively clear that the “structural fluctuation” reflected in the slope (mean square displacement, *M*) increases with temperature, the behavior shown in Figure 13 below is expected. The behavior of this figure is also in qualitative agreement with the result of the incoherent neutron scattering experiment [100].

Now, considering the Equations (27)–(31) and (34) described above, the following equation regarding the temperature dependence of the mean square displacement is obtained [2,25].
(41)$$M=k_{B}T\sum_{\alpha }^{n}\left(\frac{1}{K_{\alpha \alpha }^{E}+K_{\alpha \alpha }^{W}}\right)$$

In (41), $K_{\alpha \alpha }^{E}$ and $K_{\alpha \alpha }^{W}$ mean the “force constant” related to the restoring force of the structural fluctuation and are defined by the following equations concerning the interatomic interactions in the protein and the hydration free energy, respectively.
(42)$$K_{\alpha \alpha }^{E}=\frac{\partial^{2}U\left(\left\{R\right\}\right)}{\partial \Delta R_{\alpha }\partial \Delta R_{\alpha }}\;\;\;\;\;K_{\alpha \alpha }^{W}=\frac{\partial^{2}\Delta \mu \left(\left\{R\right\}\right)}{\partial \Delta R_{\alpha }\partial \Delta R_{\alpha }}$$

It can be readily imagined that the contribution from hydration, $K_{\alpha \alpha }^{W}$, to the force constant acts to weaken the interatomic interaction in the protein. For example, hydrogen bonds between amino acid residues or skeletal atoms in a protein are loosened or replaced by hydrogen bonds with water molecules. Then, it can be concluded that the signs of $K_{\alpha \alpha }^{E}$ and $K_{\alpha \alpha }^{W}$ must be opposite. Considering this conclusion and Equation (37), it can be concluded that the temperature dependence of the mean square displacement (*M*) of a protein in solvent is in general larger than that in a vacuum. Furthermore, it is expected that this change will occur rather abruptly at a certain temperature. This is because when the temperature of the protein solution is lowered, the fluctuation of the protein and water itself freezes, and the fluctuation is governed by the interaction energy inside the protein U({R}) in Equation (29)). It is hypothesized that the freezing temperature of this protein structure (and water) is around 230 K. In other words, it can be considered that protein fluctuations are dominated by “energy elasticity” ($K_{\alpha \alpha }^{E}$) at low temperatures, while at T<230 K by “solvent induced elasticity” ($K_{\alpha \alpha }^{W}$). This is merely a hypothesis so far, but the hypothesis is depicted conceptually in Figure 14 with corresponding experimental results by Kataoka and Nakagawa [101].

## 5. Protein Folding as an Example of Self-Organization Process

The theory of the “structural fluctuation” of protein, sketched in the previous sections, can be applied to describe the self-organization process that is one of the two elementary processes essential in life phenomena. This section is devoted to explaining an application of the theory to the *protein folding* as an example of the *self-organization processes* [103].

Under certain thermodynamic conditions (solvent environment), a protein reversibly refolds (folds) from a random coil state to its native structure (or *tertiary structure*) unique to its amino acid sequence (or *primary structure*). The process is not a spontaneous process if one focuses just on the protein structure itself, say in a vacuum, since it is a process that requires climbing up the large entropy barrier associated with ordering the structure from the disordered state of the denatured state to the native state. It is the solvent that plays a crucial role to make the process reversible. Let us quote the entire statement written by C. Anfinsen in order to explain his finding concerning the protein folding, which is called Anfinsens’s thermodynamic hypothesis [18,19]:

This hypothesis states that three-dimensional structure of a native protein in its normal physiological milieu (solvent, pH, ionic strength, presence of other components such as metal ions or prosthetic groups, temperature, and other) is the one in which the Gibbs free energy of the whole system is lowest; that is, the native conformation is determined by the totality of interatomic interactions and hence by the amino acid sequence, in a given environment [18].

Since the protein can perform its unique function only in its native conformation, the “folding” mechanism is one of the most important problems in the biophysics field as a fundamental process of life phenomenon [18,19,20,103]. Hirata and his coworkers applied the theory of “protein structure fluctuation” that has been described so far to the problem and proposed a new picture and methodology for the folding mechanism [103].

The new methodology and picture of protein folding derived from the theory described in the previous sections are as follows.

(A) Under certain thermodynamic conditions, proteins have a distribution around their equilibrium structure. The distribution is a Gaussian distribution described by Equation (31), and the variance/covariance matrix that characterizes the distribution is expressed by Equation (27).

(B) Structural changes of proteins associated with changes in thermodynamic conditions lead to changes in the mean value (equilibrium structure) and the half width (variance–covariance matrix) of this Gaussian distribution. The linear or non-linear response theory presented in Equation (34) or (35) can be applied to describe the structural change due to a thermodynamic perturbation.

The picture of protein folding deduced from Equation (35) is drawn *conceptually* in Figure 15.

Figure 15 includes two popular models of protein folding practiced in the community, the two-state model and the intermediate-state model. If the structural distribution ($\left\{R_{N}\right\}$) is a Gaussian distribution with peaks only at the two structures, the native structure ($\left\{R_{N}\right\}$) and the denatured state ($\left\{R_{N}\right\}$), as the thermodynamic condition (e.g., pressure) changes, then it can be considered as a two-state model. The intermediate state is, so to speak, a transition state, and the probability distribution of that state is so small that it can be neglected. On the other hand, if there is a significant distribution between the native structure and the denatured state, the structural transition is via the intermediate state. In order to know what kind of structural change occurs, it is necessary to solve Equation (35) by giving an atomic interaction model (or Hamiltonian) of an aqueous protein solution. In Equation (35), the perturbation ($f_{j}^{\beta }$) is an input for changing thermodynamic conditions, while the variance–covariance matrix ${\langle \Delta R\Delta R\rangle }_{\alpha \beta }$ can be calculated as the second derivative of the free energy surface with respect to the atomic coordinate of protein, based Equation (28).

## 6. Summary and Perspective

In the present article, the theoretical as well as computational studies on the molecular recognition and self-organization processes in life phenomena, carried out based on the statistical mechanics theory of solvation, or the RISM/3D-RISM theory, were reviewed. The theory provides the information concerning the position as well as orientation of small molecules around and inside protein in terms of the probability distribution of atoms, in a manner similar to the electron distribution detected by the X-ray crystallography. It was demonstrated with a few examples that the theory is able to *probe* a ligand molecule, including water, *recognized* by protein at its active site or a cavity in atomistic detail. Water molecules *recognized* by protein at its active site or a cavity are of special importance, since those water molecules play multiple roles when protein expresses its function, i.e., as substrates and nucleophiles in enzymatic hydrolysis reactions, controlling the ion mobility in an ion channel, and so on. They also make an important contribution to the binding affinity of a drug compound to a target protein, since the compound may not be accommodated at the active site of protein unless one or a few water molecules in the cavity are disposed from the cavity.

It was also clarified in the article that water plays a crucial role in the structural fluctuation of protein, which in turn makes an essential contribution to the self-organization of the biomolecule, or the *protein folding*. It can be readily imagined that a protein in a vacuum undergoes a *plastic* deformation with any finite perturbation, mechanical or thermodynamic. It is *water* that makes the protein structure *elastic*, and assists the biomolecule to refold into the native conformation. The new concept of elasticity referred to as *solvent-induced elasticity* was introduced in the article. The solvent-induced elasticity is the key to understanding why an enzyme restores its original conformation after having completed its role as a catalyst.

## Figures and Tables

**Figure 1 molecules-26-00271-f001:**
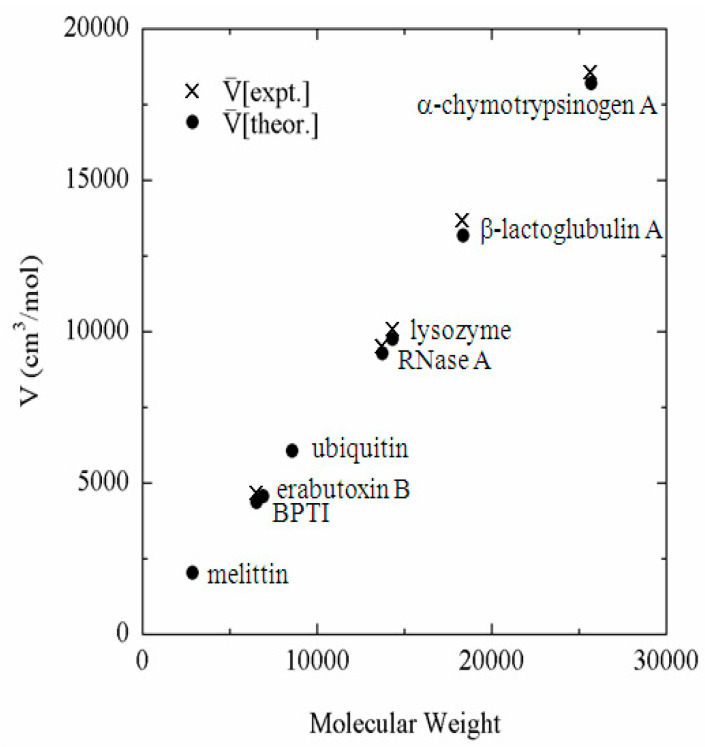
Partial molar volume of proteins in water plotted against molecular weight. (The figure was reprinted from Ref. [40]. Copyright (2005) American Chemical Society.).

**Figure 2 molecules-26-00271-f002:**
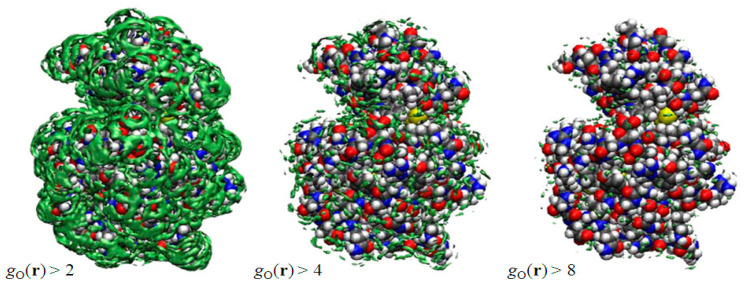
3D-distribution g(**r**) of water around a protein (lysozyme). (The figure is reprinted from Ref. [46]. Copyright (2005) American Chemical Society.)

**Figure 3 molecules-26-00271-f003:**
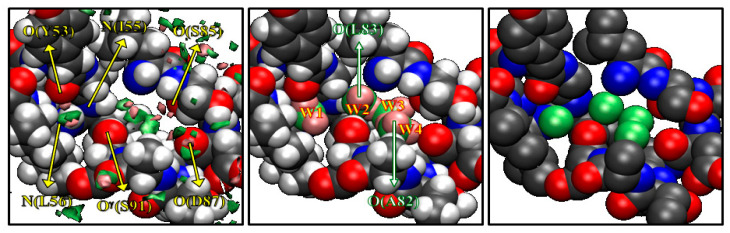
The 3D-distribution g(**r**) of water inside the active site of protein: green surface, oxygen; purple surface, hydrogen. (The figure is reprinted from Ref. [46]. Copyright (2005) American Chemical Society.)

**Figure 4 molecules-26-00271-f004:**
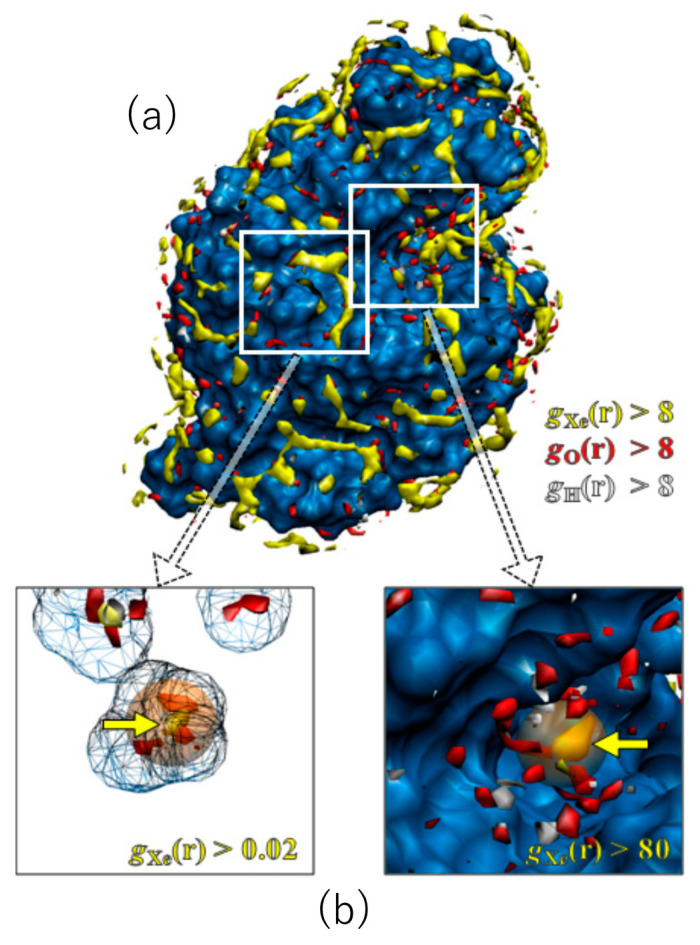
The 3D-distribution (g(**r**)) of Xe and water molecules in lysozyme: red surface, water oxygen; white surface, water hydrogen; yellow surface, xenon (**a**). The X-ray results are painted in orange (**b**). (The figure is reprinted from Ref. [54]. Copyright (2007) American Chemical Society).

**Figure 5 molecules-26-00271-f005:**
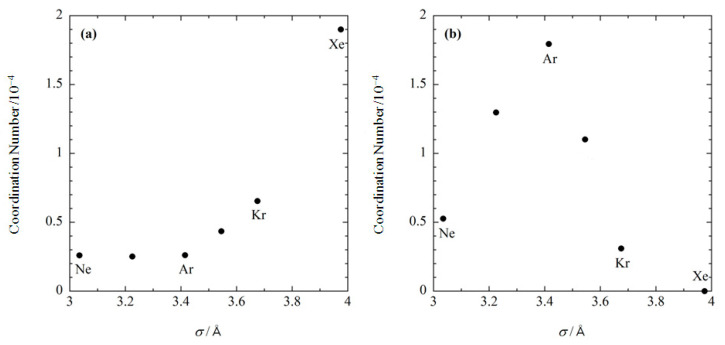
The size dependence of noble gases bound at (**a**) the substrate binding site, and (**b**) the internal site. (The figure is reprinted from Ref. [54]. Copyright (2007) American Chemical Society).

**Figure 6 molecules-26-00271-f006:**
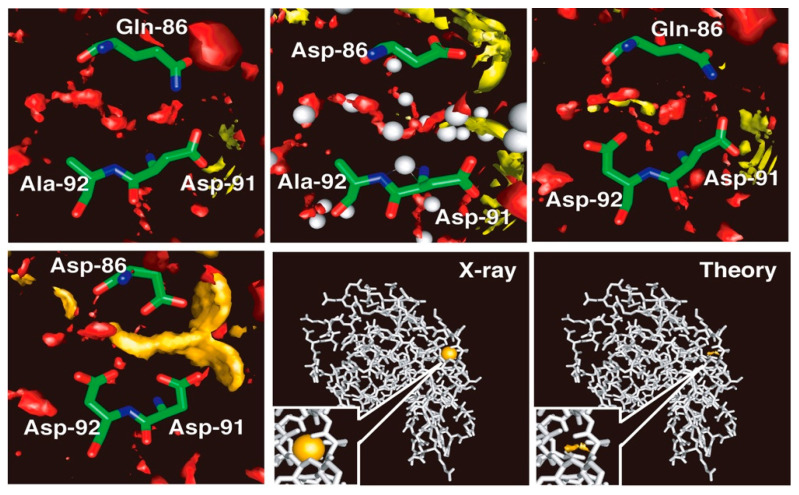
The 3D-distribution of ions in lysozyme. Upper left, wild type; upper middle, Q86D mutant; upper right, A92D mutant. lower left, Q86D /A92D mutant. The theoretical result for the Q86D /A92D mutant is compared with the result of the X-ray crystallography in the lower right two panels: The binding sites are closed up in the insets of the two panels. (The figure is reprinted from Refs. [60,61]. Copyright (2006) and (2007) American Chemical Society.)

**Figure 7 molecules-26-00271-f007:**
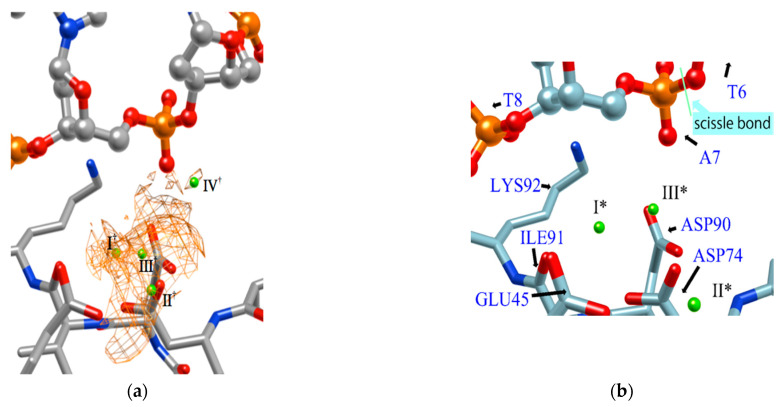
The 3D-distribution (g(**r**)) of ions in the *Eco*RV–DNA complex: (**a**) 3D-RISM/KH calculation, (**b**) Experimental results (Ref. [68]). (The figure is reprinted from Ref. [69]. Copyright (2018) American Chemical Society.).

**Figure 8 molecules-26-00271-f008:**
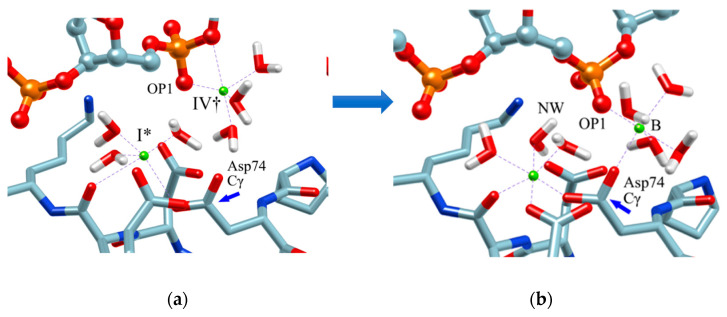
MD simulation: (**a**) initial structure, (**b**) structure after 1 nanosecond (equilibrium structure). (The figure is reprinted from Ref. [69]. Copyright (2018) American Chemical Society.)

**Figure 9 molecules-26-00271-f009:**
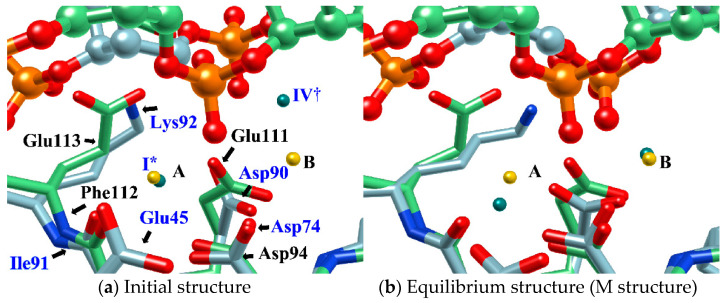
Comparison of (**a**) the MD initial structure and (**b**) the equilibrium structure (M structure) of *Eco*RV–DNA complex with *Bam*HI-DNA structure (DNA, ball and stick; protein, stick; *Eco*RV, green; *Bam*HI, blue; position of Mg^2+^ ion in MD simulation of *Eco*RV-DNA, green sphere (each label of ions in the initial structure has the same meaning as that in Figure 7); position of Ca^2+^ ion in *Bam*HI-DNA, yellow sphere). [The figure is reprinted from Ref. [69]].

**Figure 10 molecules-26-00271-f010:**
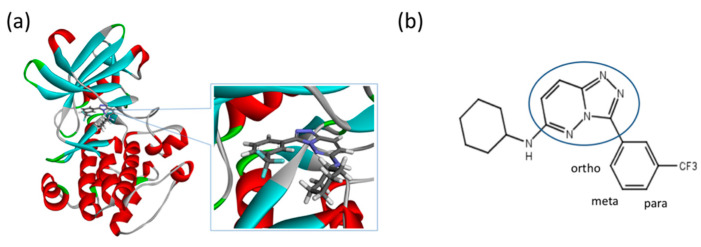
(**a**) A 3D structure of the Pim1 kinase and triazolo pyridazine inhibitor termed as VX2 in the Protein Data Bank (PDB). The PDB code is 3BGQ. We refer to the VX2 as a ligand 1, in this study, for the sake of simplicity of the terminology. (**b**) Chemical structure of the VX2. Circled part of the ligand is termed as triazolo[4,3-b]pyridazine scaffold. This is a common part of all the ligands that are applied in this study. (The figure is reprinted from Ref. [16]. Copyright (2017) American Chemical Society.)

**Figure 11 molecules-26-00271-f011:**
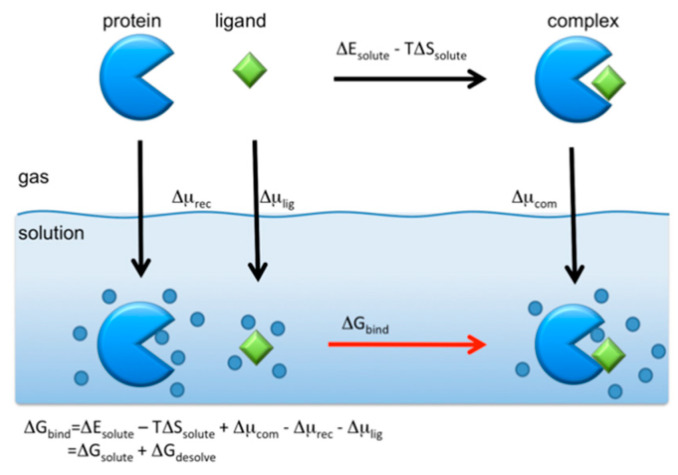
Standard thermodynamic cycle for the protein–ligand binding in aqueous solution. (The figure is reprinted from Ref. [16]. Copyright (2017) American Chemical Society.)

**Figure 12 molecules-26-00271-f012:**
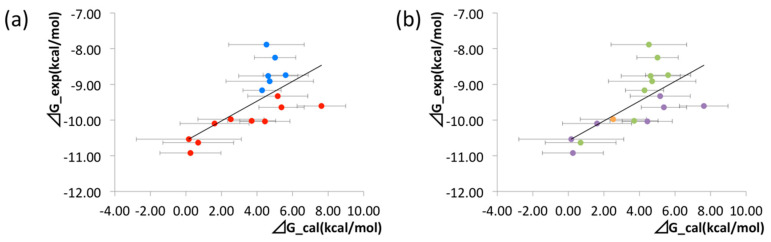
Correlation between calculated and experimental values of binding free energy for all the ligands. Correlation coefficient R = 0.69. The figures are colored in different manners based on the structural feature of the ligands. (**a**) Ligands that include CF_3_ on the meta position of the phenyl ring are colored with red. Other ligands are colored with blue. (**b**) Ligands that have cyclo-hexane, cyclo-butane, and cyclo-propane are colored with purple, orange, and green, respec-tively. (The figure is reprinted from Ref. [16]. Copyright (2017) American Chemical Society).

**Figure 13 molecules-26-00271-f013:**
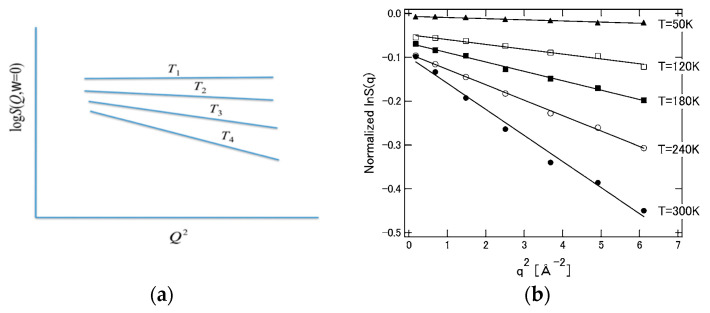
Wave number and temperature dependence of neutron structure factor. (**a**) Theoretical prediction: *T*_1_ < *T*_2_ < *T*_3_ < *T*_4_. (**b**) Experimental data Ref. [100]. ((a) is reprinted from Ref. [25]. Copyright (2018) Elsevier. (**b**) is reprinted from Ref. [100]. Copyright (2008) Elsevier.).

**Figure 14 molecules-26-00271-f014:**
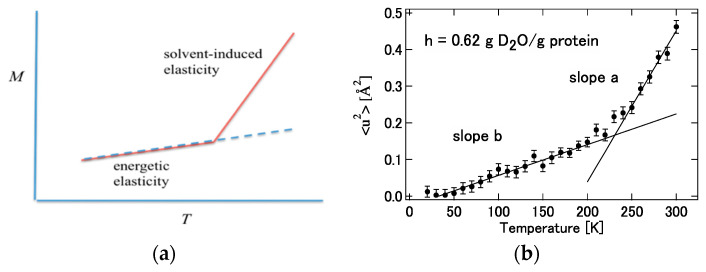
(**a**) Temperature dependence of the mean square displacement deduced from Equation (41). (**b**) Experimental results obtained by Nakagawa and Kataoka [102]. ((a) is reprinted from Ref. [25]. Copyright (2018) Elsevier. (**b**) is reprinted from Ref. [102]. Copyright (2010) Physical Society of Japan.).

**Figure 15 molecules-26-00271-f015:**
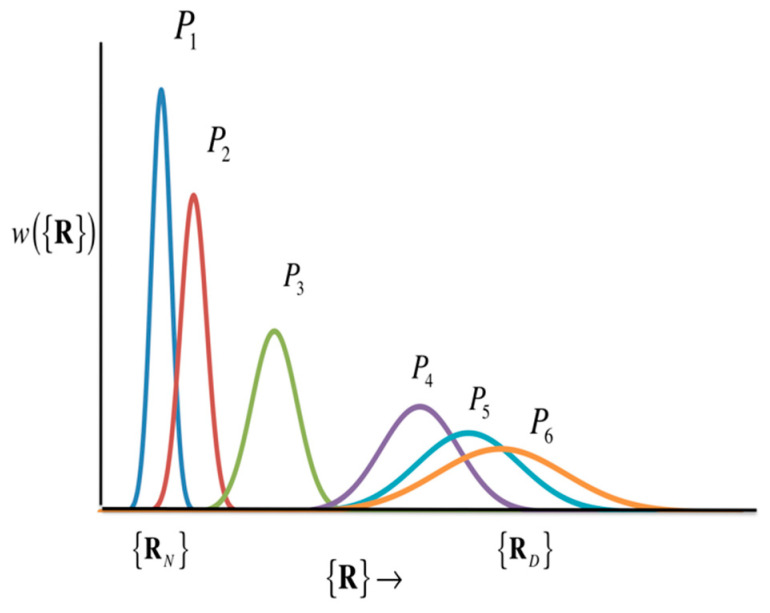
Schematic diagram showing the structural transition of proteins. (The figure is reprinted from Ref. [103]. Copyright (2018) American Institute of Physics.)

**Table 1 molecules-26-00271-t001:** The list of ligands and the inhibition constant, *K*_i._
$\Delta G_{bind,\exp}$ is the experimental value of the binding free energy estimated from -*RT*ln*K*_i_ at the temperature *T* = 300 K. $\Delta \Delta G_{bind,\exp}$ is the binding free energy relative to that of Ligand No. 1.

No.	Structure	Ki (nM)	ΔGbind,exp(kcal/mol)	ΔΔGbind,exp(kcal/mol)	No.	Structure	Ki (nM)	ΔGbind,exp(kcal/mol)	ΔΔGbind,exp(kcal/mol)
1	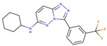	11	−10.93	0.00	9	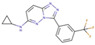	18	−10.63	0.30
2	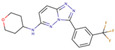	44	−10.10	0.83	10	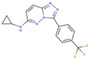	320	−8.92	2.01
3	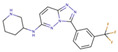	94	−9.65	1.28	11	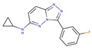	430	−8.74	2.19
4	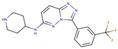	21	−10.54	0.39	12	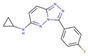	210	−9.17	1.76
5	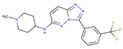	160	−9.33	1.60	13	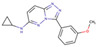	410	−8.77	2.16
6	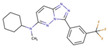	49	−10.03	0.89	14	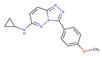	980	−8.25	2.68
7	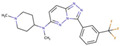	100	−9.61	1.32	15	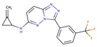	50	−10.02	0.91
8	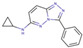	1800	−7.89	3.04	16	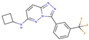	54	−9.98	0.95

**Table 2 molecules-26-00271-t002:** Binding free energy and its components obtained from MM/RISM/3D-RISM method:$\Delta E_{solute}$, the change of the interaction energy upon binding; $-T\Delta S_{solute}$, the contribution from the entropy change; $\Delta G_{solv}$, the desolvation free energy; $\Delta G_{bind\_calc}$, the theoretical estimate of the binding free energy; $\Delta G_{bind\_\exp}$, the experimental value of the binding free energy estimated from $K_{i}$ [94].

No.	$\Delta E_{solute}$ (kcal/mol)	$-T\Delta S_{solute}$ (kcal/mol)	$\Delta G_{solv}$ (kcal/mol)	$\Delta G_{bind\_calc}$ (kcal/mol)		$\Delta G_{bind\_\exp}$ (kcal/mol)
				Ave.	Std. err.	
1	−56.18	8.06	48.02	0.26	1.33	−10.93
2	−51.81	8.05	45.39	1.62	1.94	−10.10
3	−42.22	12.85	34.76	5.38	1.27	−9.65
4	−50.34	7.95	42.56	0.17	2.96	−10.54
5	−57.74	12.83	50.09	5.17	1.69	−9.33
6	−60.98	12.57	52.86	4.45	1.41	−10.03
7	−60.04	15.29	52.37	7.62	1.37	−9.61
8	−44.7	9.96	39.27	4.53	2.13	−7.89
9	−42.49	6.54	36.64	0.69	1.99	−10.63
10	−54.48	11.11	48.08	4.71	2.46	−8.92
11	−43.69	9.79	39.51	5.61	1.27	−8.74
12	−44.39	8.87	39.81	4.29	1.07	−9.17
13	−55.57	9.6	0.61	4.63	1.67	−8.77
14	−52.04	10.11	46.95	5.02	1.16	−8.25
15	−59.06	13.3	49.45	3.7	1.35	−10.02
16	−53.51	9.88	46.16	2.53	1.86	−9.98

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
