# Peer review of "Molecular Recognition and Self-Organization in Life Phenomena Studied by a Statistical Mechanics of Molecular Liquids, the RISM/3D-RISM Theory"

_molecules, 2021, doi:10.3390/molecules26020271_

Round 1
Reviewer 1 Report
The proposed work provides the review of molecular recognition and self-organization in life phenomena studied by the RISM theory. The topic itself is important and the methodology presented is potentially promising. However, the authors bring some inaccuracies in formulations which are inadmissible in the review. Also, the authors are missing important works in the field which were done by others. For example, "3D-RISM-Dock: A New Fragment-Based Drug Design Protocol" published by D. Nikolic and co-authors (J. Chem. Theory Comput. 2012, 8, 9, 3356–3372) as well as "Biomolecular Recognition Based on 3D Molecular Theory of Solvation" by the same group of authors (Biophysical Journal 106(2):411a, 2014). It appeared from the presented manuscript that everything in that area was done by one group.
Furthermore, the summary section is incomplete: "They also make an important contribution to the binding affinity of a drug compound to a target protein, since the compound may not be accommodated at the active site of protein unless one or few[???er???] water molecules in the cavity are disposed from the cavity. It was ???" (The last sentence is incomplete)
Based on the foregoing, I would suggest rejection of the article.
Author Response
(Comment by Reviewer 1)
The proposed work provides the review of molecular recognition and self-organization in life phenomena studied by the RISM theory. The topic itself is important and the methodology presented is potentially promising. However, the authors bring some inaccuracies in formulations which are inadmissible in the review. Also, the authors are missing important works in the field which were done by others. For example, "3D- RISM-Dock: A New Fragment-Based Drug Design Protocol" published by D. Nikolic and co-authors (J. Chem. Theory Comput. 2012, 8, 9, 3356–3372) as well as "Biomolecular Recognition Based on 3D Molecular Theory of Solvation" by the same group of authors (Biophysical Journal 106(2):411a, 2014). It appeared from the presented manuscript that everything in that area was done by one group.
Furthermore, the summary section is incomplete: "They also make an important contribution to the binding affinity of a drug compound to a target protein, since the compound may not be accommodated at the active site of protein unless one or few[???er???] water molecules in the cavity are disposed from the cavity. It was ???" (The last sentence is incomplete)
Based on the foregoing, I would suggest rejection of the article.
(Reply to the comment by the authors)
First of all, the authors thank to the reviewer for pointing out some errors included in the mathematical formulation of the 3D-RISM theory. In fact, we found few typographical errors in the equations, (II-14) and (II-15). Those errors are corrected in the revised manuscript.
The second point in the comment concerns the citation of the papers contributed by other authors for the application of the RISM/3D-RISM theory to the problem of the molecular recognition. The reviewer points out that some important papers are missing from the list of citation. However, the criticism is missing the main purpose of this particular article, which is described unambiguously in the abstract and introduction sections. The main purpose of the article is to clarify the physics behind the molecular recognition and the self-organization, which are the main topics of the special issue, in terms of the other physicochemical processes, the solvation and the structural fluctuation. The studies reviewed in the article are the selected topics carried out by the authors for that purpose, not for the purpose of making the exhaustive list of the papers concerning the RISM/3D-RISM theory. Nevertheless, the two papers pointed out by the reviewer are add to the list of references.
The third point made by the reviewer concerns the incomplete paragraph in the conclusion section. It was entirely a careless mistake by the authors in the file management. The manuscript is revised to be completed by adding the following paragraph.
------------------------------
It was also clarified in the article that water plays a crucial role in the structural fluctuation of protein, which in turn makes an essential contribution to the self-organization of the biomolecule, or the protein folding. It can be readily imagined that a protein in vacuum undergoes a plastic deformation with any finite perturbation, mechanical or thermodynamic. It is water that makes the protein structure elastic, and to assist the biomolecule to refold into the native conformation. The new concept of elasticity referred to as solvent-induced elasticity was introduced in the article. The solvent-induced elasticity is the key to understand why an enzyme restores its original conformation after having completed its role as a catalyst.
Reviewer 2 Report
This manuscript is a comprehensive review on summarizing the theoretical/computational studies about the molecular recognition and self-organization processes in life phenomena by Prof. Hirata's group. The paper is well-written, and I did enjoy reading paper. I would like to recommend the publication of this manuscript if the authors could consider the following minor concerns:
1) There were no references in the figure captions. Usually for review article the reference to which the figure was originally cited should be included in the figure captions.
2) I noticed some formula might not be displayed correctly: the density field equation in line 111, and the "q" variable in Eqs. IV-3 and IV7. Would a Nomenclature section in the paper be appropriate?
Reviewer 3 Report
This review covers the authors’ previous studies on molecular recognition and self-organization in life processes. Molecular recognition is the central molecular processes in life, and reviewing recent work is of value. The manuscript would be suitable for publication in Molecules once it addresses the following points:
i) A large part of the review emphasizes the role of water in molecular recognition and self-organization. As the experimental work towards elucidating the structure and function of water is challenging, various sophisticated computational methods have emerged for studying water and its role in molecular recognition. The authors describe the RISM theory, however, there are other common methods, such as WaterMap, SZMAP, WaterFLAP. The authors need to compare these computational methods in some model systems, and describe pros and cons of each of them.
ii) Desolvation of the active site is often accompanied by water reorganization in the active site upon ligand binding (cite PNAS 2011, 108, 17889-17894). This aspect is important in molecular recognition, and needs to be briefly mentioned.
iii) A well-known lock-and-key model needs to be mentioned in the introduction.
iv) Most binding work focuses on binding of ligand in the active site of proteins. The authors should briefly describe studies on mechanistically distinct allosteric binding as well.
v) References are confusing, as all give double numbers. They need to be corrected.
vi) The manuscript requires significant polishing. It is recommended that a native English speaker checks it. Some examples: … Michaelis-Menten …; Cell membranes are lipid bilayers; … ‘Protein Folding’ was first found …; … example of such process …; etc.
Author Response
(Comment by Reviewer 3)
This review covers the authors’ previous studies on molecular recognition and self-organization in life processes. Molecular recognition is the central molecular processes in life, and reviewing recent work is of value. The manuscript would be suitable for publication in Molecules once it addresses the following points:
- i) A large part of the review emphasizes the role of water in molecular recognition and self-organization. As the experimental work towards elucidating the structure and function of water is challenging, various sophisticated computational methods have emerged for studying water and its role in molecular recognition. The authors describe the RISM theory, however, there are other common methods, such as WaterMap, SZMAP, WaterFLAP. The authors need to compare these computational methods in some model systems, and describe pros and cons of each of them.
(Reply to the comment by the authors)
The main purpose of the article is to clarify the physics behind the molecular recognition and the self-organization, which are the main topics of the special issue, in terms of the other physicochemical processes, the solvation and the structural fluctuation. The studies reviewed in the article are the selected topics carried out by the authors for that purpose, not for the purpose of demonstrating the numerical accuracy of the RISM/3D-RISM theory for evaluating the thermodynamic properties of solvation. So, it is out of scope for the present review to compare the theory with the other methods such as WaterMap, SZAP, WaterFLAP for the purpose of comparing the numerical accuracy. Such work will require the benchmark type calculations concerning many different systems that are shared by those methods.
Round 2
Reviewer 1 Report
I would like to recommend the manuscript for publication.